# Aging and Cancer: The Waning of Community Bonds

**DOI:** 10.3390/cells10092269

**Published:** 2021-08-31

**Authors:** Ezio Laconi, Samuele Cheri, Maura Fanti, Fabio Marongiu

**Affiliations:** Department of Biomedical Sciences, University of Cagliari, 09124 Cagliari, Italy; cherisamuele@gmail.com (S.C.); maurafanti@usc.edu (M.F.); fabiomarongiu@unica.it (F.M.)

**Keywords:** aging, cancer, microenvironment, tissue ecology, clonal growth

## Abstract

Cancer often arises in the context of an altered tissue landscape. We argue that a major contribution of aging towards increasing the risk of neoplastic disease is conveyed through effects on the microenvironment. It is now firmly established that aged tissues are prone to develop clones of altered cells, most of which are compatible with a normal histological appearance. Such increased clonogenic potential results in part from a generalized decrease in proliferative fitness, favoring the emergence of more competitive variant clones. However, specific cellular genotypes can emerge with reduced cooperative and integrative capacity, leading to disruption of tissue architecture and paving the way towards progression to overt neoplastic phenotypes.

## 1. Aging and Cancer: The Essential Facts

Population aging is a significant aim in a society: it indicates success in the management of healthcare and social welfare but also success in the political economy, and it can be used as an index of the life quality in a nation.

On the other hand, aging is a strong risk factor for several chronic human diseases, including cancer. In a scenario where world population growth goes at the same pace as the rise of an aged group of individuals, future perspectives about incidence of neoplastic diseases are dramatically worrisome [1]. The International Agency for Research on Cancer refers to an incidence of more than 19 million cancers in the world in 2020. Only 1.4% of cancers occur before the age of 20 years, while 82% are diagnosed post 50 years. The median age of a cancer diagnosis is 66 years, and this pattern is seen in most common cancers. For example, the median age at diagnosis is 62 years for breast cancer, 67 years for colorectal cancer, 71 years for lung cancer, and 66 years for prostate cancer [2]. Furthermore, 46% of people who died from cancer worldwide in 2017 were 70 or older, and an additional 41 percent were 50 to 69 years old.

## 2. How Do We Avoid Cancer Early in Life

Given the late onset of neoplastic disease, we should acknowledge the remarkable ability of our organism to avoid the emergence of cancer during the first 40–50 years of our life. Explaining how this ability is regulated in our tissues becomes of extreme importance to comprehend its decline during aging and to devise strategies to prevent or delay such decline.

In complex organisms, such as humans, an enormous number of cells must cooperate to maintain structure and function of different tissues and organs, and to prevent or combat the emergence of non-functional/non-cooperating cells that might affect or disrupt homeostasis and eventually represent a risk for the organismal life. In order to achieve this balance, we have evolved refined and complex quality-control mechanisms for the maintenance of our tissues, aiming at maximizing the fitness of our cells while avoiding the expansion of cancerous cells. These mechanisms take into account important parameters such as the ability of cells, tissues, and organs to interact, communicate, and cooperate, while maintaining specialization and a harmonious division of labor [3,4].

Although these mechanisms have evolved to guarantee maximum efficiency during the reproductive years, thus allowing for the continuation of the species, later years in life are characterized by a decline in the effectiveness of maintenance programs [5]. Biological processes such as cell competition, cell senescence, and immune surveillance play an important role in tissue maintenance, and their decline with age might contribute to the increased risk for cancer development (Figure 1).

### 2.1. Cell Competition and the Concept of Relative Fitness

The first scientific evidence of the existence of cell competition in complex organisms was reported almost 50 years ago in a *Drosophila melanogaster* model [6]. Flies carrying a heterozygous mutation for specific ribosomal proteins (*minute*) had a normal, though slower development when in a genetically homogeneous environment (i.e., all the cells carry the same heterozygous mutation). However, when these mutations were introduced in the context of a normal tissue background in imaginal wing discs (thus generating a mosaic of *wild-type* and *minute*^+/−^ cells), mutated cells are selectively eliminated by the surrounding *wild-type* cells [6]. The paradigm here is that by comparing their relative fitness, the cell with higher fitness (*winner*) will outcompete the other (*loser*) not simply owing to a proliferative advantage but by actively eliminating the weaker counterpart [7,8]. The selection of the fittest cell will help maximize the function of a particular tissue, while preventing the expansion of potentially disruptive phenotypes.

More recently, several mutations affecting different cellular functions have been reported to alter the relative fitness of our cells compared with their *wild-type* counterparts [9]. As an example, differences in Myc levels can induce cell competition, wherein cells with higher Myc levels become “*winners*” (or supercompetitors over the *wild-type* phenotype), while cells with reduced levels of Myc are selectively eliminated [10,11,12]. This concept illustrates how, in the context of cell competition, the definition of the fitness level for a specific cell is based on the relative fitness of the surrounding cells, indicating that cells are equipped with tools for sensing the fitness levels of neighboring counterparts [9]. One of the best characterized sensors of cell fitness is the Flower protein, which, depending on the levels of expression of its three isoforms on the cell surface, can signal for the survival of the winner and/or apoptosis of the loser [13]. This winner/loser “code” has been recently reported in human cells [14].

Cell competition can therefore act as an effective barrier against the survival and expansion of pre-neoplastic cells [15,16]. In epithelial cells, a cell competition-based system has been recently described where transformed cells are actively eliminated by their healthy counterparts [17]. This has been described as an intrinsic epithelial defense against cancer (EDAC), which does not require the cooperation of the immune system [17].

Interestingly, also mutations disrupting cell polarity and tissue patterning (which could represent a trigger for cancer initiation) are considered a loser phenotype and lead to their selective elimination by the *wild-type* counterparts [18]. In the skin, for example, the active downregulation of collagen COL17A1 in damaged basal epidermal stem cells results in a reduced formation of hemidesmosomes with the basement membrane, leading to the extrusion of these cells from the skin epithelium and the preservation of tissue homeostasis [19].

It is important to underline that, in order to maintain overall size and function, our tissues have the hierarchic ability to control the rate of proliferation of winner cells and/or elimination of losers. Therefore, in the context of cell competition, the elimination of a damaged cell only occurs when a fitter cell is available to replace the one that is lost [20].

Thanks to evolutionary processes, during the reproductive years, our tissues are equipped with cells at a near-optimal fitness level. Therefore, the most likely scenario is that any insult to our cells will result in a decreased fitness and the elimination of damaged cells via cell competition, so long as fitter cells are available to replace them. However, over time, with the progressive accumulation of damage in a growing number of cells, the ability of our tissues to eliminate loser cells will decrease as the fitness of neighboring cells is also reduced. This will increase the chances for mutations to confer a higher fitness and to lead to the emergence and proliferation of oncogenic clones [5]. For example, DNA damage response activity deteriorates with age, likely resulting in age-related functional p53 loss and leading to accumulated DNA damage and chromosomal instability [21].

### 2.2. Cell Senescence

Another important strategy to maintain our tissues and prevent the expansion of altered cells is the induction of cell senescence. When cells are faced with structural damage or dysregulated growth signals, they can either undergo apoptosis or enter a state of persistent cell cycle arrest, without immediate cell death (i.e., cell senescence) [22]. This mitotic block represents an effective means to prevent the clonal expansion of DNA-damaged cells at risk for oncogenic initiation [23]. Nonetheless, senescent cells are metabolically active and retain cell-type specific functions. Moreover, they secrete an array of signals (e.g., pro-inflammatory cytokines, growth- and matrix-remodeling factors) that can have profound effects on the surrounding microenvironment (e.g., fueling low grade inflammation [24,25]). While these signals are generally beneficial and promote tissue regeneration in the short term, the persistence of senescence cells within our tissues and their progressive accumulation with aging can be detrimental, contributing to tissue dysfunction and fueling carcinogenesis through different mechanisms [26,27].

Although senescent cells can be eliminated by both innate and adaptive immune-mediated mechanisms [28], during aging we observe a decline of this clearing capacity [29].

### 2.3. Immune Surveillance

Our immune system represents one of the most important mechanisms of quality control in our cells and tissues, acting as an effective barrier to cancer development [30,31]. Both innate and adaptive immune cells can actively eliminate damaged or altered cells through different mechanisms [32]. As an example, NK cells can recognize specific ligands that are overexpressed on the surface of senescent cells [33]. Additionally, adaptive T cell-mediated immune responses can target specific neoantigens expressed by malignant cells, thus contributing to their elimination [34].

While very effective in keeping cancer at bay during our reproductive years, the immune system itself undergoes a progressive decline with aging, contributing to the establishment of the so-called “immunosenescence” [35,36]. For instance, aging is characterized by a decline in macrophage metabolic and immune function, with a reduced clearance and immunosurveillance capacity [37]. Senescent macrophages in old mice were shown to contribute to the low-grade systemic inflammation commonly referred to as *inflammaging* [38]. Older individuals are also equipped with a reduced number of tissue-resident antigen presenting cells with a limited capacity to migrate to secondary lymphoid tissues and stimulate T cells activation [39]. The effectiveness of adaptive immunity is also intrinsically dampened with old age. T cell response undergoes major age-dependent changes that gradually compromise its main functionality. Thymic involution, mitochondrial dysfunction, genetic and epigenetic alterations, loss of proteostasis, and eventually senescence have all been reported to affect T lymphocyte function, and consequently, the proper activation of a complete adaptive response [35].

## 3. Aging and Cancer: How Does It Happen

### 3.1. A Cell-Oriented View

The idea that the origin of cancer can be largely (if not exclusively) explained through (genetic) alterations occurring in rare cells undergoing neoplastic transformation is still a dominant one in the field [40,41]. In essence, this line of thought, often referred to as the somatic mutation theory, posits that the neoplastic phenotype results from the progressive accumulation of critical mutagenic events in target cells and that this is both necessary and sufficient to drive their invasive and metastatic behavior [42,43,44]. Such an assumption also informs most of the current approaches of targeted cancer therapies aimed at countering specific molecular pathways associated with the mutant genotype [45]. Within this perspective, the most direct mechanistic hypothesis that attempts to account for the link between aging and neoplastic disease is based, inferentially, on the increased likelihood for mutagenic events to accumulate in cells as the individual ages [41,46,47]. Effective anti-oxidant defenses and DNA repair pathways, together with the elimination of genetically damaged cells via differentiation, cell competition, and/or immune-mediated clearance (Figure 1, see also preceding paragraphs), largely reduce the possibility for the accrual of DNA damage in aged tissues [46,47,48,49]. However, none of these protective strategies attains perfect efficiency, thereby leading to the accumulation of DNA alterations that are typically observed during aging in several cell types, including stem cell compartments [40,46,47,48,49]. Building on the somatic mutation theory of cancer development, it is then postulated that the age-associated progressive rise in mutational burden increases the possibility of the appearance of overtly neoplastic cells endowed with the right combination of altered genes [50,51,52]. A more updated/refined version of this hypothesis centers on the emergence of rare cells harboring a “mutator phenotype”, which would set the stage for additional genetic alterations and neoplastic progression [53]. While this type of scenario may capture important biological attributes of cancer cell populations, such as genetic instability, it places emphasis only on events taking place within cells undergoing transformation. By contrast, it completely overlooks any possible pathogenic role of age-associated changes—such as mutagenic events occurring in the bulk of the tissue and/or organism—towrds explaining the link between aging and cancer [54,55].

### 3.2. A Tissue-Oriented View

It is axiomatic that the aging process entails a host of complex changes at both the structural and functional level affecting every cell, tissue, and organ in the body [56]. On the other hand, studies conducted over the past several decades have unequivocally established the fundamental involvement of the surrounding environment in the origin and progression of neoplastic disease [54,57,58,59,60]. Thus, an important goal of cancer research is to elucidate to what extent age-related alterations in the tissue landscape could account for the increased risk of cancer in the older population. In one of the first accounts on this topic, McCullough et al. showed that the liver microenvironment of the aged rat is more permissive for the growth of transplanted neoplastic epithelial cells compared with that of young recipients [61]. Subsequently, results obtained by our research group indicated that the hepatic milieu of old animals is also clonogenic for normal transplanted hepatocytes, in that cells isolated from normal donors and infused into young or old syngeneic host livers formed larger cluster in the latter compared with the former [62]. Furthermore, a similar age-related differential response in growth rate was observed following orthotopic transplantation of pre-neoplastic cells isolated from chemically induced hepatocyte nodules [63]. Meanwhile, an increasing number of reports have suggested a mechanistic link between aging microenvironments and neoplastic disease in several tissues, including bone marrow [64], lung [65], skin [66], colon [59], prostate [67], ovary [68], and mammary gland, among others. However, the biological bases for this association have yet to be fully elucidated. Widespread, low grade inflammation, which is of common occurrence during aging [69], has been often implicated as being responsible, at least in part, for the increased burden of neoplastic disease in the old [70,71,72], in continuity with Virchow’s irritation theory of the origin of cancer [73]. Local production of oxidative species, tissue damage, and regeneration driven by cytokines and growth factors secreted by inflammatory cells represent possible mediators of this effect [74,75], although their specific role remains to be established, if any. Aging-related fibrosis and the resulting increase in tissue stiffness have also been proposed to fuel carcinogenesis [76,77,78,79], possibly via alterations in the mechanical force balance between ECM, cell, and cytoskeleton [80]. However, it should be pointed out that neoplastic disease generally begins as a focal lesion originating from rare cells undergoing clonal expansion, implying that widespread alterations in the tissue microenvironment can only act as selective forces to favor the emergence of those cells endowed with specific (advantageous) phenotypes.

### 3.3. An Ecological View

The latter consideration calls for a more comprehensive and ecological view regarding the relationship between aging, altered microenvironment, and risk of neoplastic disease [5,81]. As early as in 1938, the Scottish pathologist Alexander Haddow acutely pointed out that, since carcinogens are generally inhibitors of cell proliferation, their pro-neoplastic effect could only result from the emergence of cells expressing a resistant phenotype in response to such proliferative constraint [82], thereby proposing a *bona fide* ecological interpretation of cancer development. About 40 years later, selection of preneoplastic/resistant cells in the context of a growth-suppressed tissue environment was formally demonstrated as a mechanism able to fuel carcinogenesis [83], a principle that was later confirmed by other studies [60,84].

Following these observations, it was proposed that a similar paradigm could possibly apply to the alterations occurring in normal tissues as a consequence of the aging process [85]. Given that random mutations result in the formation of predominantly deleterious allele [1,46,47,86,87], their accumulation in cells during aging will translate into a decrease in functional proficiency, including proliferative and regenerative capacity [88,89]. For example, liver regeneration is delayed in elderly patients [90] and experimental animals [88], and this is due, at least in part, to a cell-autonomous decrease in proliferative potential of the aged hepatocyte [91]. Moreover, it adds to the observation referred to above that the aged liver microenvironment is clonogenic to both normal [62] and pre-neoplastic [63] transplanted hepatocytes. It is reasonable to suggest that such increased clonogenicity might be related to an overall decrease in regenerative potential of the bulk of the tissue that selects for cells with advantageous genotypes/phenotypes [92]. An intriguing report on the effect of smoking on human bronchial mucosa supports this postulation. The majority of bronchial epithelial cells exposed to smoking suffer a high mutational burden, as expected. However, it was found that a relatively rare population of mitotically quiescent cells are relatively spared of mutagenic events and are able to repopulate the mucosal epithelium upon smoking cessation [93], implying that relatively normal (wild-type) cells are able to out-compete heavily mutagenized populations.

The clonogenic/selective property of the aged tissue microenvironment has emphatically emerged in more recent years, following a series of reports documenting the common presence of aberrant clonal expansions in normal tissues of old individuals [94]. This phenomenon was initially described in bone marrow-derived cell populations [95] and later extended to solid organs such as skin [96], esophagus [97], and liver [98], among others. In fact, it was found that aged tissues, albeit histologically normal, are often populated by a patchwork of mutant clones that appear to be positively selected under the specific (micro)environmental conditions associated with aging [99]. Thus, the human esophageal epithelium is progressively colonized by clonal populations carrying alterations in several genes, with a predominant presence of Notch1 mutants [97,100]. Interestingly, the size of these clones was further increased in smokers and in alcohol drinkers (56), indicating that these toxic exposures, in synergy with the aging process, favor the expansion of such altered cellular genotypes. This in turn would suggest that their emergence in old individuals is similarly fueled by widespread damage accumulating in tissues as we age, leading to selection of clones that have gained a competitive advantage.

The above evidence is consistent with the hypothesis that the pervasive presence of mutant clones in tissues of old individuals can be ascribed to the inherent clonogenic potential of the aged tissue microenvironment resulting, at least in part, from a widespread age-associated decline in cell and tissue functional fitness, including proliferative fitness [64,91]. Under these conditions, rare clones, including clones bearing mutations that have been linked to neoplastic transformation, may gain an edge over surrounding homotypic cells, thereby undergoing selective expansion. Notably, such competitive advantage does not necessarily entail a mutant phenotype. In fact, normal (young) hepatocytes form large clusters upon transplantation in the liver of an old (but not young!) recipient [62,101]; similarly, bronchial epithelial cells that were relatively shielded from the mutagenic effect of cigarette smoke were able to outcompete surrounding mucosal cells that were heavily hit by genotoxic damage, as mentioned above [93]. It is also implied in this interpretation that such clonal expansions are part of the normal cell turnover in that tissue and therefore occur, at least initially, within the limits imposed by homeostatic control mechanisms [102]. However, they pose an overall increased risk of neoplastic progression, and this is likely to depend on the specific cellular phenotype that has been selected [54]. Along these lines, we have provided evidence to suggest that a focal growth pattern, rather than clonal growth per se, represents a critical hallmark of pre-neoplastic lesions, including polyps, nodules/adenomas, and papillomas, while clones that are histologically normal and well-integrated in the host tissue bear little or no relevance to neoplastic disease [103,104]. Within this perspective, cancer is fundamentally interpreted as a disease originating from an alteration in tissue pattern formation [105,106,107].

## 4. Aging and Cancer: How to Loosen the Link

Given the strong association between aging and neoplastic disease, two questions become relevant in order to implement preventive strategies to attenuate the impact of cancer on the human population. First, it is important to learn whether major avoidable risk factors, such as smoking and UV light, increase the risk of cancer via mechanisms that are shared, at least in part, with the process of aging. Second, although chronological aging cannot be averted, a viable option would be to delay biological alterations associated with the process that are mechanistically related to the origin of cancer, thereby retarding the emergence of neoplastic disease.

### 4.1. Cancer Risk Factors and Aging

Insights into the former question have come from elegant studies conducted on esophageal epithelium of transgenic mice. Under normal conditions, rare p53 mutant cells are present in the mucosal progenitor cell compartment; exposure to low-dose ionizing radiation (LDIR) blocks proliferation of wild-type progenitor cells and drives their differentiation towards upper layers, thereby providing a competitive advantage to cells expressing the p53-altered genotype [108]. Hence, clones of p53 mutants emerge throughout the mucosa. However, when LDIR-induced oxidative damage was prevented with an antioxidant, wild-type progenitor cells were no longer out-competed and were able to keep at bay p53 mutants [108]. A similar sequence of events has been proposed to occur following exposure to UV light in human skin [109], suggesting that the clonogenic capacity of radiation is related, at least in part, to detrimental effects exerted on the bulk of the tissue, as proposed for the aging process. Analogies between photoaging and intrinsic skin aging have recently been highlighted, with specific reference to their possible role in the pathogenesis of melanoma [66].

Another case in point pertains to the effect of smoking on bronchial epithelium referred to above [93]. While mutant clones affected 4–14% of bronchial mucosal cells in middle-aged, non-smoking individuals, such proportion was increased to at least 25% in current smokers, implying that smoking adds to a process of clonal growth that is also associated with aging. Concurrently, smoking increased overall mutational burden, as expected, contributing a total of 1000 to 10,000 additional mutations per cell [93]. Thus, it appears that the emergence of altered clones associated with smoking occurs in the context of widespread toxicity exerted on the tissue landscape [85].

In summary, both intrinsic aging and exposure to major exogenous risk factors for human neoplasia, such as radiation (UV light) and smoking, result in chronic, cumulative damage to the target tissue, which in turn selects for the growth of clones with the fittest phenotype. While the process per se is adaptive in nature [81,85], it can also set the stage for subsequent steps of neoplastic evolution [102].

### 4.2. Dietary Interventions

Prevention of cancer is largely based on avoidance of risk factors. For example, a decrease in smoking translates to a sizeable decline in lung cancer incidence and mortality, as observed over the last few decades in Western countries [110]. From this perspective, the risk of neoplasia associated with aging is certainly difficult to tackle, since aging, at least with reference the time component, is not something anyone would like to avoid. However, the evidence discussed in the preceding paragraphs points to biological alterations associated with aging as playing a role in the pathogenesis of neoplastic disease, as opposed to chronological aging per se. Within this framework, it becomes reasonable to conceive possible strategies aimed at slowing down and/or delaying the onset of those specific changes that render the aged tissue more prone to cancer development. Among such strategies, dietary interventions have been revealed to be particularly effective [111].

Starting from over a century ago, numerous reports have consistently indicated that caloric restriction (CR), i.e., a reduction in caloric intake compared with ad libitum feeding without causing malnutrition, translates into a decrease in the rate of biological aging, with a parallel reduction in the incidence of age-related diseases, including cancer ([74] and references therein). In more recent years, these observations have been extended to non-human primates [112,113]. At the metabolic/molecular level, CR is able to modulate several biochemical pathways, including mTOR, which has been implicated in its beneficial effects on healthspan and lifespan [114].

However impressive as these results may be, their translation to the human experience has been difficult to implement, mainly, albeit not exclusively [115], because of socio-cultural reasons. Thus, alternative approaches have been pursued, focusing on protocols that reproduce, as far as possible, the beneficial effects of CR while avoiding those that are less acceptable or appealing to humans. They can be broadly grouped into the following: (i) time-restricted feeding/eating (TRF/E), which involves a limited time-window of food consumption (usually 8–12 h/day); (ii) intermittent fasting (IF), consisting of periodic fasting intervals (2–3/week),with a minimum length of 18 h; (iii) fast-mimicking diet (FMD), based on cycles of low caloric intake (500–1000 kcal/day, for 4–5 days) that are reiterated every two weeks, monthly, or bimonthly. As it is readily evident, a unifying feature of these approaches is the presence of a fasting period. This may not be surprising since a long fasting interval is also included in CR protocols, given that experimental animals exposed to CR typically consume their food ration within <8 h [116,117]. Data are now emerging that these dietary interventions can exert a beneficial effect on carcinogenesis in experimental settings. For example, TRF, similar to CR, was able to delay the growth of transplanted preneoplastic hepatocytes in the liver of aged rats, possibly via effects on the tissue microenvironment [116,117]. Furthermore, progression of spontaneous lymphomas was retarded in mice exposed to FMD [118] or to alternate-day fasting [119]. In an intriguing recent development, FMD has been shown to potentiate the efficacy of chemotherapy and hormonal therapy in patients with breast cancer [120], suggesting that dietary approaches can exert profound effects on aging and age-associated diseases at various stages of these processes, including the latest stages.

## 5. Conclusions

While the intricacies of the relationship between aging and cancer are far from being completely unveiled, important tiles of the mosaic begin to emerge. Based on current evidence, an ecological view encompassing both the relevance of specific (rare) altered cellular genotypes and the instructive role of age-induced changes in the tissue context stands as the most comprehensive approach to this complex issue (Figure 2).

Given the reported high frequency of altered cellular genotypes in tissues of normal individuals at relatively young age [121], the rate limiting step for the actual formation of mutant clones seems to reside largely in the emergence of a selective (clonogenic) tissue microenvironment, which is typically associated with older age [5,40]. The majority of such clones bear no direct relevance to neoplastic development and are rather interpretable as an expression of normal cell turnover regulated by competing phenotypes, with no signs of altered tissue architecture and/or growth autonomy. In fact, as the burden of damage (mutational or otherwise, e.g., non-enzymatic glycation) increases with age, average cellular fitness decreases [64,91] and becomes more heterogeneous, resulting in winner and loser phenotypes. On the other hand, specific genetic alterations may confer both a competitive advantage in proliferative fitness and properties that are disruptive of tissue integrity: as discussed above, the resulting clones with a focal growth pattern might pose a risk for evolution towards neoplasia [105,122]. Thus, it would appear that neither an altered genotype alone nor an altered tissue context alone are sufficient to drive carcinogenesis, at least in the large majority of cancer cases associated with advanced age (Figure 2).

Multicellular organisms are based on an overarching, unifying principle holding together a society of mutually collaborative cells. It is obvious that such a complex community is held together through a continuous flow of information and feedback mechanisms prioritizing the common good at the expense of over-competitive behaviors. Aging also entails a progressive waning of this principle, and individual cells are more and more likely to be trapped in a “break the lines” message, a siren’s song with catastrophic consequences.

## Figures and Tables

**Figure 1 cells-10-02269-f001:**
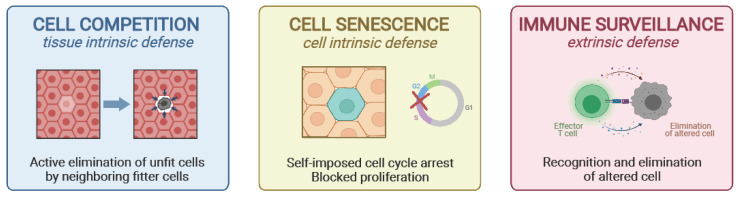
Protective mechanisms against cancer. The survival and proliferation of altered cells can be prevented by three main mechanisms. Cell competition (**left panel**) is a tissue intrinsic mechanism, where unfit cells can be actively eliminated by fitter surrounding cells; cell senescence (**central panel**) is a cell intrinsic mechanism, where altered cells enter a permanent state of mitotic block; immune surveillance (**right panel**) represents an important tissue extrinsic mechanism for the elimination of senescent or otherwise altered cells.

**Figure 2 cells-10-02269-f002:**
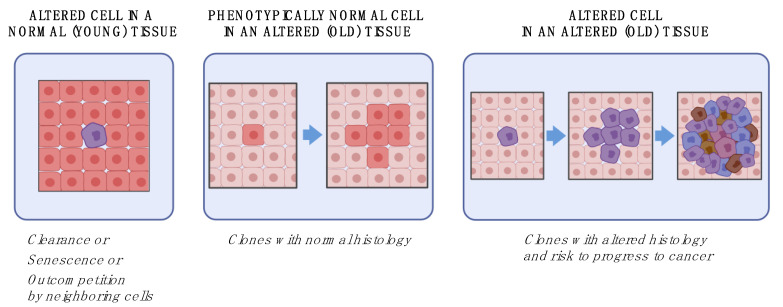
Aging and cancer: an ecological view. If an altered cell emerges in a normal tissue environment (**left panel**), it will be most likely cleared or kept at bay by mechanisms outlined in Figure 1. On the other hand, widespread alterations in the tissue environment, such as during aging (**central panel**), can select for the emergence of more fit, phenotypically normal cells, forming histologically normal clones. The combined presence of altered cellular phenotypes in an altered tissue environment (**right panel**) can set the stage for the evolution of neoplastic disease.

## Data Availability

Data supporting reported results can be found on PubMed.

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
