# Peer review of "Aging and Cancer: The Waning of Community Bonds"

_cells, 2021, doi:10.3390/cells10092269_

Round 1

Reviewer 1 Report

original and very relevant review article on the link between cancer and aging.

Very well written.

I only have a minor remark; they could also brielfy mention in chapter 4 the role of mTOR on the aging process (which is probably linked to the effect seen with dietary interventions).

Author Response

I only have a minor remark; they could also briefly mention in chapter 4 the role of mTOR on the aging process (which is probably linked to the effect seen with dietary interventions).

This is a pertinent suggestion. The possible role of mTOR was briefly addressed in the revised version.

Reviewer 2 Report

Laconi and co-authors introduce an interesting and relatively underappreciated idea that the local tissue microenvironment sets the stage for cancer development, especially in the setting of aging tissue. They postulate that in normal tissue, mechanisms such as cell competition, senescence, and immune surveillance supervise and suppress the expansion of mutated clones. They also suggest that tissue structure changes with age due to wear and tear, leading to accumulated DNA damage and weakened protective mechanisms, and that these tissue alterations create a microenvironment permissive for clonal neoplastic development.

The major concern is that this review is lacking information on molecular mechanisms altering age-related normal tissue microenvironment. Here are only few examples of possible mechanisms that have been discussed in the literature:

  • DNA damage response activity deteriorates with age, likely resulting in age-related functional p53 loss and leading to accumulated DNA damage and chromosomal instability (Feng et al, PNAS 2007; Gutierrez-Martinez et al, Nat Cell Biol 2018).
  • Senescent cells release senescence associated secretory phenotype (SASP), which can significantly change the tissue microenvironment, inducing senescence in neighboring cells and creating low-grade tissue inflammation (Acosta et al, Nat Cell Biol 2013).

In addition, the authors describe dietary interventions that have anti-aging effects without mentioning molecular mechanisms underlying these effects in tissue, including metabolic changes. Furthermore, age-related deficits of some hormones (e.g., estrogen, growth hormone) and excess of others (e.g., glucocorticoids) may also alter normal tissue microenvironment and these are not discussed.

Laconi and co-authors introduce an interesting and relatively underappreciated idea that the local tissue microenvironment sets the stage for cancer development, especially in the setting of aging tissue. They postulate that in normal tissue, mechanisms such as cell competition, senescence, and immune surveillance supervise and suppress the expansion of mutated clones. They also suggest that tissue structure changes with age due to wear and tear, leading to accumulated DNA damage and weakened protective mechanisms, and that these tissue alterations create a microenvironment permissive for clonal neoplastic development.

The major concern is that this review is lacking information on molecular mechanisms altering age-related normal tissue microenvironment. Here are only few examples of possible mechanisms that have been discussed in the literature:

  • DNA damage response activity deteriorates with age, likely resulting in age-related functional p53 loss and leading to accumulated DNA damage and chromosomal instability (Feng et al, PNAS 2007; Gutierrez-Martinez et al, Nat Cell Biol 2018).
  • Senescent cells release senescence associated secretory phenotype (SASP), which can significantly change the tissue microenvironment, inducing senescence in neighboring cells and creating low-grade tissue inflammation (Acosta et al, Nat Cell Biol 2013).

In addition, the authors describe dietary interventions that have anti-aging effects without mentioning molecular mechanisms underlying these effects in tissue, including metabolic changes. Furthermore, age-related deficits of some hormones (e.g., estrogen, growth hormone) and excess of others (e.g., glucocorticoids) may also alter normal tissue microenvironment and these are not discussed.

Laconi and co-authors introduce an interesting and relatively underappreciated idea that the local tissue microenvironment sets the stage for cancer development, especially in the setting of aging tissue. They postulate that in normal tissue, mechanisms such as cell competition, senescence, and immune surveillance supervise and suppress the expansion of mutated clones. They also suggest that tissue structure changes with age due to wear and tear, leading to accumulated DNA damage and weakened protective mechanisms, and that these tissue alterations create a microenvironment permissive for clonal neoplastic development.

The major concern is that this review is lacking information on molecular mechanisms altering age-related normal tissue microenvironment. Here are only few examples of possible mechanisms that have been discussed in the literature:

  • DNA damage response activity deteriorates with age, likely resulting in age-related functional p53 loss and leading to accumulated DNA damage and chromosomal instability (Feng et al, PNAS 2007; Gutierrez-Martinez et al, Nat Cell Biol 2018).
  • Senescent cells release senescence associated secretory phenotype (SASP), which can significantly change the tissue microenvironment, inducing senescence in neighboring cells and creating low-grade tissue inflammation (Acosta et al, Nat Cell Biol 2013).

In addition, the authors describe dietary interventions that have anti-aging effects without mentioning molecular mechanisms underlying these effects in tissue, including metabolic changes. Furthermore, age-related deficits of some hormones (e.g., estrogen, growth hormone) and excess of others (e.g., glucocorticoids) may also alter normal tissue microenvironment and these are not discussed.

Laconi and co-authors introduce an interesting and relatively underappreciated idea that the local tissue microenvironment sets the stage for cancer development, especially in the setting of aging tissue. They postulate that in normal tissue, mechanisms such as cell competition, senescence, and immune surveillance supervise and suppress the expansion of mutated clones. They also suggest that tissue structure changes with age due to wear and tear, leading to accumulated DNA damage and weakened protective mechanisms, and that these tissue alterations create a microenvironment permissive for clonal neoplastic development.

The major concern is that this review is lacking information on molecular mechanisms altering age-related normal tissue microenvironment. Here are only few examples of possible mechanisms that have been discussed in the literature:

  • DNA damage response activity deteriorates with age, likely resulting in age-related functional p53 loss and leading to accumulated DNA damage and chromosomal instability (Feng et al, PNAS 2007; Gutierrez-Martinez et al, Nat Cell Biol 2018).
  • Senescent cells release senescence associated secretory phenotype (SASP), which can significantly change the tissue microenvironment, inducing senescence in neighboring cells and creating low-grade tissue inflammation (Acosta et al, Nat Cell Biol 2013).

In addition, the authors describe dietary interventions that have anti-aging effects without mentioning molecular mechanisms underlying these effects in tissue, including metabolic changes. Furthermore, age-related deficits of some hormones (e.g., estrogen, growth hormone) and excess of others (e.g., glucocorticoids) may also alter normal tissue microenvironment and these are not discussed.

Laconi and co-authors introduce an interesting and relatively underappreciated idea that the local tissue microenvironment sets the stage for cancer development, especially in the setting of aging tissue. They postulate that in normal tissue, mechanisms such as cell competition, senescence, and immune surveillance supervise and suppress the expansion of mutated clones. They also suggest that tissue structure changes with age due to wear and tear, leading to accumulated DNA damage and weakened protective mechanisms, and that these tissue alterations create a microenvironment permissive for clonal neoplastic development.

The major concern is that this review is lacking information on molecular mechanisms altering age-related normal tissue microenvironment. Here are only few examples of possible mechanisms that have been discussed in the literature:

  • DNA damage response activity deteriorates with age, likely resulting in age-related functional p53 loss and leading to accumulated DNA damage and chromosomal instability (Feng et al, PNAS 2007; Gutierrez-Martinez et al, Nat Cell Biol 2018).
  • Senescent cells release senescence associated secretory phenotype (SASP), which can significantly change the tissue microenvironment, inducing senescence in neighboring cells and creating low-grade tissue inflammation (Acosta et al, Nat Cell Biol 2013).

In addition, the authors describe dietary interventions that have anti-aging effects without mentioning molecular mechanisms underlying these effects in tissue, including metabolic changes. Furthermore, age-related deficits of some hormones (e.g., estrogen, growth hormone) and excess of others (e.g., glucocorticoids) may also alter normal tissue microenvironment and these are not discussed.

Author Response

The major concern is that this review is lacking information on molecular mechanisms altering age-related normal tissue microenvironment. Here are only few examples of possible mechanisms that have been discussed in the literature:

DNA damage response activity deteriorates with age, likely resulting in age-related functional p53 loss and leading to accumulated DNA damage and chromosomal instability (Feng et al, PNAS 2007; Gutierrez-Martinez et al, Nat Cell Biol 2018).

Senescent cells release senescence associated secretory phenotype (SASP), which can significantly change the tissue microenvironment, inducing senescence in neighboring cells and creating low-grade tissue inflammation (Acosta et al, Nat Cell Biol 2013).

In addition, the authors describe dietary interventions that have anti-aging effects without mentioning molecular mechanisms underlying these effects in tissue, including metabolic changes. Furthermore, age-related deficits of some hormones (e.g., estrogen, growth hormone) and excess of others (e.g., glucocorticoids) may also alter normal tissue microenvironment and these are not discussed.

We thank the reviewer for raising these important issues, which were addressed in the revised version of the MS.

Round 2

Reviewer 2 Report

no comments